# Synthesis and Characterization of Zinc Oxide Nanoparticles at Different pH Values from *Clinopodium vulgare* L. and Their Assessment as an Antimicrobial Agent and Biomedical Application

**DOI:** 10.3390/mi14071285

**Published:** 2023-06-23

**Authors:** Hajira Arif, Sidra Qayyum, Wasim Akhtar, Iram Fatima, Waqas Khan Kayani, Khursheed Ur Rahman, Wedad A. Al-Onazi, Amal M. Al-Mohaimeed, Naila Khan Bangash, Nasra Ashraf, Sarah Abdul Razak, Asif Kamal, Sajid Ali

**Affiliations:** 1Department of Botany, University of Azad Jammu and Kashmir, Muzaffarabad 13100, Pakistan; hajra.12@gmail.com (H.A.); sidraqayum1@gmail.com (S.Q.); 2Department of Biotechnology, Fatima Jinnah Women University, Rawalpindi 46000, Pakistan; iramfatima46@yahoo.com; 3Department of Biotechnology, Faculty of Basic and Applied Sciences, University of Kotli, Azad Jammu and Kashmir, Kotli 11100, Pakistan; wkkayani@gmail.com; 4Department of Botany, Hazara University, Mansehra 21120, Pakistan; khursheed823@yahoo.com; 5Department of Chemistry, College of Science, King Saud University, P.O. Box 22452, Riyadh 11495, Saudi Arabia; walonzai@ksu.edu.sa (W.A.A.-O.); muhemeed@ksu.edu.sa (A.M.A.-M.); 6Environmental Microbiology Laboratory, Department of Agricultural Biological Chemistry, College of Agriculture and Life Sciences, Chonnam National University, Gwangju 61186, Republic of Korea; naila.quaidian@gmail.com; 7Department of Zoology, University of Azad Jammu and Kashmir, Muzaffarabad 13100, Pakistan; n.ashraf@gmail.com; 8Institute of Biological Sciences, Faculty of Science, Universiti Malaya, Kuala Lumpur 50603, Malaysia; sarahrazak@um.edu.my; 9Department of Plant Sciences, Faculty of Biological Sciences, Quaid-i-Azam University, Islamabad 45320, Pakistan; a.kamal@bs.qau.edu.pk; 10Department of Horticulture and Life Science, Yeungnam University, Gyeongsan 38541, Republic of Korea

**Keywords:** antimicrobial assay, *Clinopodium vulgare*, biomedical, pH values, zinc oxide

## Abstract

The current study attempts to evaluate the formation, morphology, and physico-chemical properties of zinc oxide nanoparticles (ZnO NPs) synthesized from *Clinopodium vulgare* extract at different pH values and to investigate their antimicrobial and biomedical application potential. The reduction of zinc ions to ZnO NPs was determined by UV spectra, which revealed absorption peaks at 390 nm at pH 5 and 348 nm at pH 9, respectively. The spherical morphology of the nanoparticles was observed using scanning electron microscopy (SEM), and the size was 47 nm for pH 5 and 45 nm for pH 9. Fourier-transformed infrared spectroscopy (FTIR) was used to reveal the presence of functional groups on the surface of nanoparticles. The antibacterial activity was examined against *Staphylococcus aureus*, *Streptococcus pyogenes*, and *Klebsiella pneumonia* via the agar-well diffusion method. Comparatively, the highest activities were recorded at pH 9 against all bacterial strains, and among these, biogenic ZnO NPs displayed the maximum inhibition zone (i.e., 20.88 ± 0.79 mm) against *S. aureus*. ZnO NPs prepared at pH 9 exhibited the highest antifungal activity of 80% at 25 mg/mL and antileishmanial activity of 82% at 400 mg/mL. Altogether, ZnO NPs synthesized at pH 9 show promising antimicrobial potential and could be used for biomedical applications.

## 1. Introduction

Nanotechnology has made major developments in the biogenic synthesis of nanoparticles (NPs) that can be applied in biomedical, agricultural, and chemical engineering fields [1]. Traditional chemical and physical methods to produce NPs resulted in some toxic chemicals being adsorbed on the surface during their synthesis, which causes negative effects in biological applications of nanoparticles [2]. Non-toxic and environmentally friendly biosynthesis of metal NPs is now the target of recent research applications due to their toxicity-reducing ability and antimicrobial potential [3]. Further, the physio-chemical properties of NPs obtained after green synthesis have the added benefit of increasing the life span of NPs in comparison to traditional chemical and physical methods [4].

Metal oxide NPs are now used in the biomedical field due to their large surface-to-volume ratio, as well as in industrial applications [5]. Metal and metal oxide NPs are found to be antioxidants and antibacterial in practical applications [6]. Low concentrations of metal oxide NPs, including zinc (Zn), magnesium (Mg), and titanium (Ti) oxides, inhibit microbial growth significantly [7]. Moreover, the metal NPs can be generated using a green strategy that employs various plant components as reducing agents. The NPs synthesized via the green approach are biocompatible, cost-effective, and less toxic [8]. Among metal oxides, ZnO has attracted great interest owing to its unique characteristics, such as its high photochemical and catalytic activity [9]. The synthesis and nature of NPs can be observed by changing the experiment’s pH, as it is one of the important factors influencing the ZnO properties. It also affects the hydrolysis and condensation behavior of the solution and the size of NPs. Therefore, pH values directly impact the morphology and biological properties of ZnO NPs.

Herbal products are enriched with pharmacologically active compounds. Since ancient times, plants have been utilized for diverse purposes, including medicines, dyes, cosmetics, disinfectants, and industrial applications [10]. *Clinopodium vulgare* L. (Lamiaceae), commonly known as wild basil, is a well-known medicinal plant used in traditional medicines worldwide. *C. vulgare* has a wide range of ethnopharmacological uses, making it a popular choice for treating various conditions such as hemorrhagic disease, ulcers, diabetes, mastitis, prostatitis, and skin inflammation [11]. According to previous reports, *C. vulgare* contains phenolic, flavonoid, alkaloid, and terpenoids that act as reducing agents and possess antimicrobial activities [12,13].

To scientifically validate the traditional medicinal applications of *C. vulgare*, a comprehensive study was designed so that the plant extracts could be exploited as reducing and capping agents for the green synthesis of ZnONPs through zinc salt solutions. The green synthesis of ZnONPs was evaluated using *C. vulgare* at high and low pH, as literature on the synthesis, characterization, and biological properties of ZnO NPs at different pH values is lacking. Hence, the NPs were assessed at different pH values to better understand the differences in the physio-chemical properties of NPs obtained through green synthesis. Moreover, their antimicrobial and antileishmanial properties were examined for the first time.

## 2. Materials and Methods

### 2.1. Extract Preparation

*C. vulgare* (L.) leaves were collected from different areas of the district of Muzaffarabad. The raw material used was pure, as it was verified by the taxonomist to avoid any ambiguity. The leaves were thoroughly washed with tap water to remove dust particles and then dried in shade for almost 2 weeks. 10 g of leaf powder was mixed with 100 mL of distilled water and heated for 30 min at 60 °C. The solution was cooled down, followed by filtration using Whattman filter paper, and then the extract was modified separately by adding acid (HCl) for acidic pH 5 and base (NaOH) for basic pH 9. Subsequently, the extract was preserved at 4 °C for the synthesis of NPs.

### 2.2. Synthesis of ZnONPs

Initially, 100 mM of zinc sulfate (ZnSO_4_) monohydrate salt concentration was used in 500 mL of distilled water with 5 mL of 1 M sodium hydroxide. Subsequently, the mixture was stirred for 30 min and placed in an amber-colored container to prevent auto-oxidation.

ZnO NPs were prepared from *C. vulgare* extract under specific pH 5 and pH 9 conditions. Briefly, 100 mL of *C. vulgare* extracts were added to 500 mL of a 100 mM zinc oxide solution separately. After setting the pH, both solutions were heated for 20 min at 60 °C and allowed to stand at room temperature. The color change from light brown to pale yellow was noted upon ZnONP synthesis reaction completion. Eventually, the mixture was centrifuged for 30 min at 12,000 rpm to settle down the synthesized NPs, and the pellet was washed three times with autoclaved distilled water before being dried overnight in the oven [14].

### 2.3. Characterization of ZnO NPs

The biogenic ZnONPs were characterized using various analytical techniques. The UV absorption spectrum was evaluated at wavelengths ranging from 100 to 800 nm using a Perkin Elmer Lambda 950 UV/Vis Spectrophotometer. The crystal nature was observed via X-ray diffraction analysis (XRD). The morphology and appearance of ZnONPs were observed via scanning electron microscopy (SEM), energy dispersive X-ray analysis (EDX) was applied for elemental analysis, and fourier-transform infrared spectroscopy (FTIR; Shimadzu-8400S) was used to observe the presence of different functional groups at the NPs surfaces.

### 2.4. Antibacterial Assay

The antibacterial action of synthesized ZnONPs was assessed using the agar-well diffusion method [15]. For bacterial culture, the nutrient broth media and Mueller–Hinton agar were used, and the activity was tested against both gram-positive (*Staphylococcus aureus* and *Streptococcus pyogenes*) and gram-negative (*Klebsiella pneumonia*) bacteria. Different concentrations of ZnONPs (50–400 µg/mL) along with an antibiotic (Clindamycin) as a positive control were used over agar plates. After 24 h, the inhibition zones were measured with the help of a ruler.

### 2.5. Antifungal Activity

For antifungal analysis, in vitro ZnO NPs prepared in different concentrations (25–100 mg/mL) were added separately to PDA media and three replicates of each concentration were made. Fungal strains were used for this assay. The 4 nm inoculum discs of isolated fungi were excised with the help of a cork borer and placed in the center of PDA media plates in sterilized conditions. A PDA media plate without nanoparticles served as a positive control, and inoculated petri plates were kept in an incubator at 25 ± 2 °C to grow. After one week, growth inhibition at different concentrations of nanoparticles in each petri plate was calculated by the following formula:Growth inhibition % = (C − T)/C × 100(1)
where C denotes fungal growth in control plate and T shows fungal growth in treated plates.

### 2.6. Antileishmanial Activity

The samples were tested for leishmanicidal activity using the well-known leishmanial parasite *Leishmania tropica* (KWH23), following the standard protocol [16]. The assay was performed on both the promastigote (flagellar) and amastigote (aflagellate) forms. Newly formed promastigotes and amastigotes were placed in MI99 media with 10% FBS. In a separate 96-well plate, 20 mL of the test sample and 180 mL of the refreshed culture were gently combined at the time of the assay. After three days, the plates were re-incubated for 72 h at 25 °C. Amphotericin-B was used as a positive control (PC) in the experiment, while DMSO was taken as a negative control (NC). Following the initial incubation period, each well of the plate was filled with 20 mL of MTT solution (4 mg/mL DW) and left at room temperature for another incubation. The absorbance of the plate was measured at 540 nm, and the percentage of inhibition was calculated using the equation below.
(%) Inhibition = 100 × Absample/Abcontrol(2)

### 2.7. Statistics

The biological assay was performed in triplicate, and mean values were measured. SPSS software was used for the statistical analysis. The standard deviation from absolute data was assessed as a measure of dispersion, and the distribution of data was expressed using the mean and standard deviation in a tabulated form.

## 3. Results and Discussions

### 3.1. Biogenic Synthesis of ZnONPs

In the present study, NPs were synthesized at pH 5 and pH 9 using *C. vulgare* extract as a reducing and stabilizing agent. Different characterization techniques were conducted to ascertain the formation of ZnO NPs. Initially, the color change from dark brown to pale yellow indicated the synthesis of ZnO NPs (Figure 1). The synthesis was found to be more successful at pH 9.

### 3.2. Characterization of ZnO NPs

#### 3.2.1. UV-Spectra Analysis

After the reduction process, the green synthesis of ZnONPs was confirmed by observing the respective absorption spectra at 100 to 800 nm wavelength. The UV-Vis spectra of synthesized ZnO NPs are presented in Figure 2a,b. At pH 5, the UV spectra showed an absorption peak at 390 nm, while at pH 9, the absorption peak was noted at 348 nm. At pH 9, the synthesis was noted earlier than at pH 5. Moreover, the formation of NPs was highly detected at 9 pH. The present study correlates with previous studies in which researchers confirmed the absorption peaks at 340–390 nm for ZnONPs [17].

#### 3.2.2. X-ray Diffraction Analysis

Further studies were committed by using XRD to explain the nature of nanoparticles (Figure 3a,b). The peak position was consistent with metallic ZnO NPs. This technique was based on projecting a monochromatic X-ray beam onto the material. Bragg reflection with 2θ values for 56.8°, 62.9°, 64.05°, and 66.14° and 30°, 36°, 39°, 47°, 55°, 63°, and 68° were obtained at pH 5 and 9, respectively, and the Scherrer equation was used to calculate the size of NPs. The strong peak indicates the ZnO-NPs the planes of the XRD pattern have good agreement with (JCPDS card no. 5-0664). The average calculated diameter of nanoparticles is between 47 and 45 nm at pH 5 and pH 9. Other bilges showed impurities such as the phyto-chemicals used in the synthesis of NPs [18].

#### 3.2.3. SEM Analysis

SEM was used for the morphological description of ZnONPs, which revealed that the appearance was less spherical at pH 5 (Figure 4a) while becoming reduced and more circular at pH 9 (Figure 4b). The size ranges from 47 nm for pH 5 to 45 nm for pH 9. These results also agree with the previous reports [19,20]. The surface area and biological activities tend to increase with the decrease in size of NPs [18]. Thus, it can be suggested that NP synthesis at pH 9 has a significant yield of harvesting. For confirmation of elemental analysis, EDX was performed (Figure 5a,b). The EDX results confirmed the presence of ZnO, and these findings are similar to previously documented results [21].

#### 3.2.4. FTIR Spectroscopy Analysis

FTIR was used to recognize various chemical functional groups associated with the synthesis of ZnONPs at pH 5 and 9. The spectra confirmed that the reduction that occurred in zinc oxide ions to zinc oxide nanoparticles was due to the reducing substance in plant extract. The peaks centered at 3385 cm^−1^, 3279 cm^−1^, 2920 cm^−1^, and 2850 cm^−1^ indicated O–H and C–H stretching. Peaks located at 2375 cm^−1^, 1740 cm^−1^, and 1655 cm^−1^ correspond to C–H, C=O, and C=C bends. Similarly, peaks at 1596 cm^−1^, 1420 cm^−1^, 1080 cm^−1^, and 630 cm^−1^ signify N–H, C–C, C–N, and C–Cl/C–Br bonds from biogenic ZnO NPs (Figure 6a,b). The FTIR of only plants was also performed (Figure 7).

Previously, researchers also showed different functional groups indicating phenolic, flavonoid, and terpenoid compounds that play key roles in the formation and stabilization of NPs [22,23]. In the present study, shifts in peaks indicated the presence of different functional groups in *C. vulgare* extract that are involved in the synthesis of ZnO NPs and prevent agglomeration at different pH values [24]. The biomolecules in the plant extract serve as capping agents and stabilize NPs by various mechanisms such as electrostatic stability, hydration force stabilization, and van der Waals forces. The NP’s stability is significant for its functions and applications [25].

### 3.3. Antibacterial Assay

Antibacterial activity of *C. vulgare*-mediated ZnO NPs was tested against three bacterial strains at three different concentrations with clindamycin as a standard. The highest anti-bacterial activity of biogenic ZnONPs was noted with the maximum inhibition zone (i.e., 20.88 ± 0.79 mm) against *S. aureus* at pH 9. While the minimum inhibition zone (i.e., 10.33 ± 0.48 mm) was observed against *K. pneumonia* at pH 5. It shows the possible potential of biogenically synthesized ZnONPs at high pH. Overall, the activity was found to be concentration-dependent against all bacterial strains (Table 1).

The present study showed the significant antibacterial potential of bio-synthesized ZnoNPs from *C. vulgare* plant extracts [26,27]. The increased antibacterial activity of ZnO NPs can be attributed to the bioactive functional groups existing on the surface of NPs [28]. Reactive oxygen species (ROS) generation is the key mechanism that gives antibacterial potential to NPs [29,30]. Moreover, membrane protein damage due to NPs adsorption on the surface results in bacterial cell damage. The surface defect in the symmetry of NPs causes bacterial inhibition and injury to cells [31]. Furthermore, various functional groups occurring in *C. vulgare* extract resulted in capped ZnO NPs, which play a significant role in bacterial inhibition.

Overall, the ZnO NPs prepared at pH 9 showed potential activity in comparison to those prepared at pH 5, so the NPs prepared at pH 9 were used for further activities.

### 3.4. Antifungal Assay

One of the great benefits of biosynthesized zinc oxide nanoparticles against pathogenic fungi was investigated. Different concentrations of nanoparticles (prepared at pH 9) including 25 mg/mL, 50 mg/mL, 75 mg/mL, and 100 mg/mL were used to check the inhibition effect on PDA media. Results of the antifungal activity revealed significant growth inhibition at 25 mg/mL (80%), 50 mg/mL (75%), 75mg/L (68%), and 100 mg/mL (59%) concentrations of ZnO NPs (Figure 8). Our findings have also shown that a lower concentration of ZnO NPs is more effective in disease control, and the nanoparticles have a significant role in disease regulation in an eco-friendly way. In the previous few years, scientists have effectively used ZnO nanocomposite to stop the growth of various fungi like *Candida albicans*, *Fusarium graminearum*, and *Aspergillus niger* [32,33,34]. The results of the antifungal potential of ZnO NPs follow a similar pattern to previous research [35,36,37].

### 3.5. Antileishmanial Activity

The green synthesized ZnO NPs were tested for their antileishmanial activity against leishmaniasis, an infectious disease primarily caused by the parasite Leishmania. Leishmaniasis has an annual incidence rate of 1.5 to 2 million worldwide, despite being endemic in 100 nations. In humans, sand flies transmit the intracellular leishmania parasite, and this disease is spread by the biting parasites Phlebotomus and Lutzomyia. Due to the wrong vectors, this disease is very likely to spread uncontrollably. The cytotoxicity of formulations of synthesized Zn ONPs was tested on amastigote and *L. tropica* promastigote cultures at concentrations ranging from 50 to 400 g/mL. As can be seen, cytotoxicity at various doses was found to be effective against the parasite’s promastigote and amastigote forms, with mortality rates of 77% 1.4 and 74% 1.6 at 400 g/mL (Table 2). Furthermore, both patristic forms had significant LC 50 values of 220 g/mL for promastigote (Phlebotomus and Lutzomyia) bites and 270 g/mL for amastigote (Table 3). All these are aligned with the previous reports [38,39,40].

## 4. Conclusions

It is known that the green synthesis of ZnONPs is much safer and more environmentally friendly as compared to chemical synthesis. The green synthesis of nanoparticles is a cost-effective and eco-friendly method. The synthesized ZnO NPs were assessed through various techniques, including spectroscopic and microscopic analysis, including XRD, FTIR, UV, SEM, and EDX. The role of the bioactive compounds in *C. vulgare* leaves as an oxidizing agent has been discussed. The antibacterial, antifungal, and antileishmanial potentials were observed to be significantly higher at pH 9, indicating that alkaline medium (pH 9) is best suitable for NP synthesis in comparison to acidic medium (pH 5). The results of this work highlight the potential of *C. vulgare* leaves to synthesize multifunctional ZnO NPs used for antimicrobial activities. Zinc oxide nanoparticles synthesized from the green synthesis pathway are expected to have more extensive applications in biotechnology, sensors, medical, catalysis, optical devices, coatings, and drug delivery.

## Figures and Tables

**Figure 1 micromachines-14-01285-f001:**
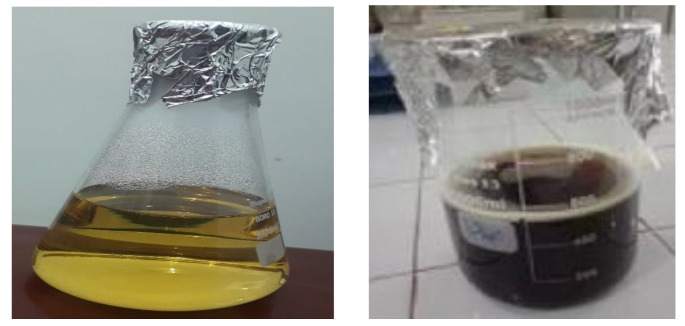
The change in color of the solution from dark brown to pale yellow observed after adding zinc sulfate.

**Figure 2 micromachines-14-01285-f002:**
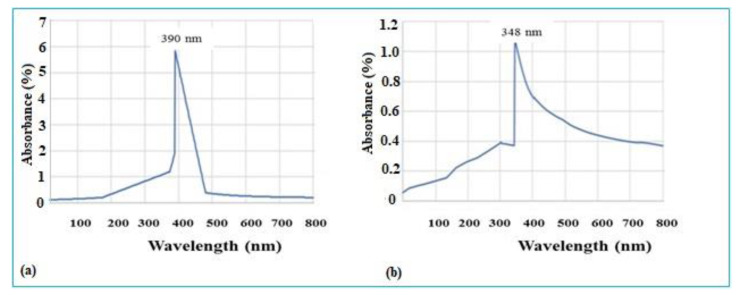
UV-Vis spectra showing the absorption wavelengths of as-synthesized biogenic ZnO NPs (**a**) at pH 5 and (**b**) at pH 9.

**Figure 3 micromachines-14-01285-f003:**
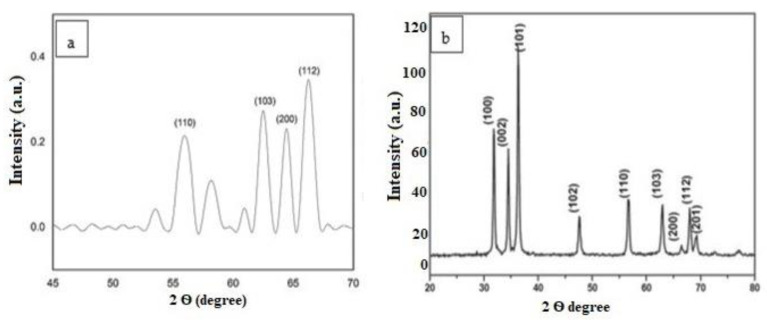
XRD showing the absorption wavelengths of biogenic ZnO NPs (**a**) at pH 5 and (**b**) at pH 9.

**Figure 4 micromachines-14-01285-f004:**
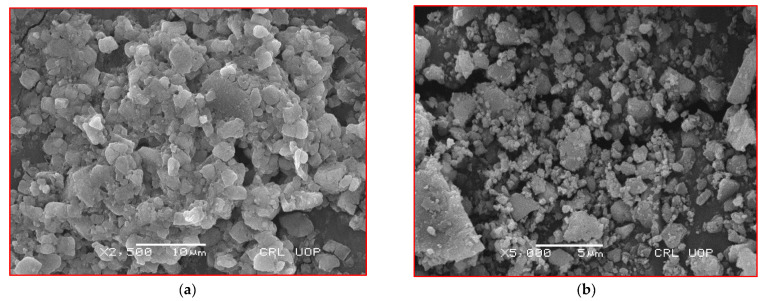
SEM images of *C. vulgare*-mediated ZnONPs using ZnSO_4_ salt at various pH values (**a**) at pH 5 and (**b**) at pH 9.

**Figure 5 micromachines-14-01285-f005:**
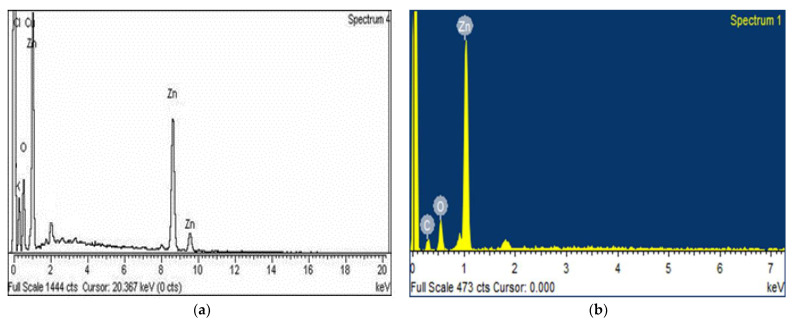
EDX spectra of *C. vulgare* mediated ZnONPs using ZnSO_4_ salt at various pH values (**a**) at pH 5 and (**b**) at pH 9.

**Figure 6 micromachines-14-01285-f006:**
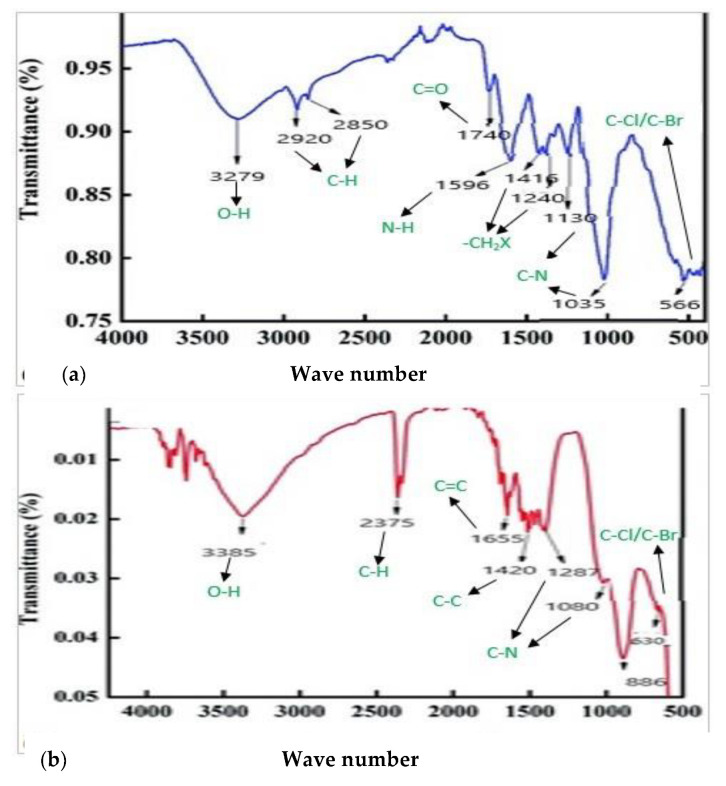
FTIR spectra of *C. vulgare* mediated ZnONPs (**a**) at pH 5 and (**b**) at pH 9.

**Figure 7 micromachines-14-01285-f007:**
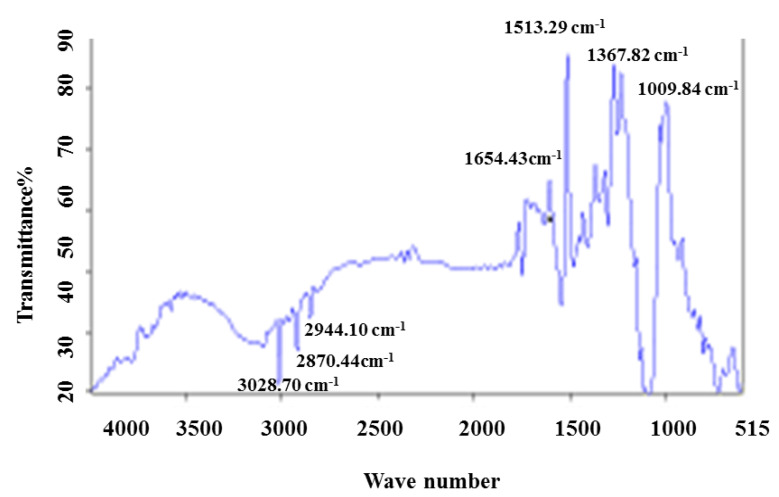
FTIR spectra of *C. vulgare* plant extract.

**Figure 8 micromachines-14-01285-f008:**
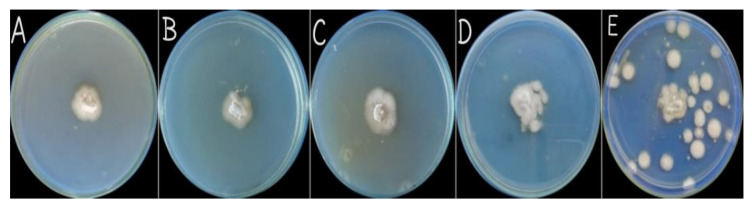
Growth inhibition of pathogenic fungi at different concentrations 25 mg/mL (**A**), 50 mg/mL (**B**), 75 mg/mL (**C**), 100 mg/mL (**D**), and 0 mg/mL (**E**) of zinc oxide nanoparticles.

**Table 1 micromachines-14-01285-t001:** Antibacterial activities of biogenic ZnONPs determined at different concentrations.

pH	Bacterial Pathogens	Inhibition Zones Observed at Different Concentrations (mm ± SD)			Standard (Clindamycin)
		5 µg/mL	10 µg/mL	15 µg/mL	
5	*Streptococcus pyogenes*	10.33 ± 0.57	11.66 ± 0.57	13.33 ± 0.57	18.00 ± 0.54
	*Staphylococcus aureus*	12.62 ± 1.15	14.66 ± 0.57	16.66 ± 0.57	19.01 ± 0.54
	*Klebsiella pneumonia*	8.33 ± 0.47	8.66 ± 0.47	10.33 ± 0.48	17.02 ± 0.57
9	*Streptococcus pyogenes*	12.60 ± 0.57	14.60 ± 0.58	16.60 ± 0.59	18.22 ± 0.54
	*Staphylococcus aureus*	16.33 ± 0.57	18.66 ± 0.59	20.88 ± 0.79	19.11 ± 0.54
	*Klebsiella pneumonia*	8.66 ± 0.47	10.33 ± 0.47	12.66 ± 0.47	17.33 ± 0.57

**Table 2 micromachines-14-01285-t002:** Comparative analysis of prepared nanoparticles with previous work.

S. No	Type of Nanoparticles	Source	Antimicrobial Potential	Reference
1.	MB-ZnO NPs	Plants	60%	Huang, P., et al., 2020 [41]
2.	ZnO NPs	Mushroom	70%	Hayat et al., 2022 [42]
3.	ZnO NPs	Chemical synthesized	76%	Dehghani et al., 2019 [43]
4.	Zeolite/ZnO	Physiochemical method	87%	Yang et al., 2022 [44]
5.	ZnO	Plant extract	80% (antifungal)	Current work
6.	ZnO	Plant extract	82% (Antileishmanial)	Current work

**Table 3 micromachines-14-01285-t003:** Antileishmanial activity of synthesized ZnO NP.

Conc. (ug/mL)	Promastigote Inhibition	Amastigote	IC50 (ug/mL)	IC50 (ug/mL)
400	82.40	75.15	270 Amastigote	220 Promastigote
200	69.90	63.00		
100	57.13	54.22		
50	41.20	38.20		

## Data Availability

Not applicable.

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
