# Peer review of "Synthesis and Characterization of Zinc Oxide Nanoparticles at Different pH Values from Clinopodium vulgare L. and Their Assessment as an Antimicrobial Agent and Biomedical Application"

_micromachines, 2023, doi:10.3390/mi14071285_

Round 1

Reviewer 1 Report

Journal: Micromachines

Manuscript ID: micromachines-2444706

Title: Synthesis, and Characterization, of Zinc Oxide nanoparticles at different pH values from Clinopodium vulgare L. and their assessment as an antimicrobial agent and biomedical application

Arif et al. attempted to evaluate the formation, morphology, and physico-chemical properties of zinc oxide nanoparticles (ZnO NPs) synthesized from Clinopodium vulgare extract at different pH values. The additional aim was to investigate ZnO NPs' antimicrobial and biomedical application potential. This topic is interesting, but the manuscript is poorly prepared. My specific comments are given below.

-Author Asif Kamal is listed two times in the author list.

-The Introduction is rather short but informative. The novelty and the motivation for the presented manuscript should be clearly stated at the end of the Introduction.

-Line 92: N or M? Please, revise.

-Line 95: the authors did not prepare extract in the range of pH but at pH 5 and 9. Please, revise.

-Line 97: heated shortly? Please, define the temperature and time.

-Lines 138 and 139: Please, revise. The sentence is confusing.

-Line 152: pH 5 and 9, not the whole range.

-Line 160: the authors claim that they observed UV Vis spectrum in the range of 100 – 900nm, but the specification for the instrument they use specifies the wavelength range from 175. Also, from the presented spectra, it is obvious that their upper limit was 800 nm.

-Line 164: Please, revise.

-Figure 2: y-axis should be labeled properly. X-axis cannot start from 0. The sharpness of the picks presented seems strange. Please, explain.

-Line 177: Please, revise.

-Figure 3: The resolution of the figure is poor. Figures are not uniform. The axis should be labeled properly. The title of the figure should be revised.

-Figure 4: The scale is missing. –Figure 5: Please, explain the presence of Cu in EDX. Figures are not uniform, and the resolution is poor.

-Line 228: Please, revise.

-Figures 6 and 7: Resolution is horrible, and axis labeling is unacceptable.

-Section 3.3: What about the influence of pH and the extracts? Please, discuss.

-Line 270: if it is on the surface, it is adsorption, not absorption.

-The explanation for the exclusion of ZnO NPs on pH 5 from further investigation is not evident.

-Lines 279-280: please, revise.

-Line 281: How did the authors calculate NPs concentration?

-the term "mycosinthesized" should be reconsidered.

-Line 312: NPs instead of Nps.

-Line 313: Please, revise.  

Minor editing of English language is required. 

Author Response

Reviewer 1

Arif et al. attempted to evaluate the formation, morphology, and physico-chemical properties of zinc oxide nanoparticles (ZnO NPs) synthesized from Clinopodium vulgare extract at different pH values. The additional aim was to investigate ZnO NPs' antimicrobial and biomedical application potential. This topic is interesting, but the manuscript is poorly prepared. My specific comments are given below.

Response: Thank you so much sir. The manuscript has been revised. The quality has been improved.

Query: Author Asif Kamal is listed two times in the author list.

Response: Correction has been done.

Query: The Introduction is rather short but informative. The novelty and the motivation for the presented manuscript should be clearly stated at the end of the Introduction.

Response: Suggested change has been incorporated.

Query: Line 92: N or M? Please, revise.

Response: Correction has been done.

Query: Line 95: the authors did not prepare extract in the range of pH but at pH 5 and 9. Please, revise.

Response: Correction has been done.

Query: Line 97: heated shortly? Please, define the temperature and time.

Response: Correction has been done.

Query: Lines 138 and 139: Please, revise. The sentence is confusing.

Response: The line has been revised.

Query: Line 152: pH 5 and 9, not the whole range.

Response: Done

Query: Line 160: the authors claim that they observed UV Vis spectrum in the range of 100 – 900nm, but the specification for the instrument they use specifies the wavelength range from 175. Also, from the presented spectra, it is obvious that their upper limit was 800 nm.

Response: UV range has been corrected.

Query: Line 164: Please, revise.

Response: The line has been revised.

Query: Figure 2: y-axis should be labeled properly. X-axis cannot start from 0. The sharpness of the picks presented seems strange. Please, explain.

Response: Suggested change has been done.

Query: Line 177: Please, revise.

Response: The line has been revised.

Query: Figure 3: The resolution of the figure is poor. Figures are not uniform. The axis should be labeled properly. The title of the figure should be revised.

Response: Correction has been done.

Query: Figure 4: The scale is missing. –Figure 5: Please, explain the presence of Cu in EDX. Figures are not uniform, and the resolution is poor.

Response: Scale has been added in Figure 4. The presence of Cu in EDX confirms nanoparticle preparation via the green synthesis method. The Cu may come from the plant extract which was present as a raw material.The figures have been changed as per suggestion.

Query: Line 228: Please, revise.

Response: The line has been revised.

Figures 6 and 7: Resolution is horrible, and axis labeling is unacceptable.

Response: The figures has been modified to improve the resolution.

Query: Section 3.3: What about the influence of pH and the extracts? Please, discuss.

Response: Influence of pH have been elaborated now.

Query: Line 270: if it is on the surface, it is adsorption, not absorption.

Response: Correction has be done

The explanation for the exclusion of ZnO NPs on pH 5 from further investigation is not evident.

Response: Added on page 12.

Query: Lines 279-280: please, revise.

Response: Done

Query: Line 281: How did the authors calculate NPs concentration?

Response: The nanoparticles concentration was used according to previous reported literature.

Query: the term "mycosinthesized" should be reconsidered.

Response: Instead of mycosinthesized the term green synthesized was used.

Query: Line 312: NPs instead of Nps.

Response: Done

Query: Line 313: Please, revise.  

Response: Done

Query: The quality of the figures in general are bad, they should be improved.

Response: Figures have been modified as per suggestion.

Reviewer 2 Report

Upon reviewing your manuscript intitled: Synthesis, and Characterization, of Zinc Oxide nanoparticles at different pH values from Clinopodium vulgare L. and their assessment as an antimicrobial agent and biomedical application. I find your work interesting, but I do not believe it can be published in its current form. Therefore, some revisions must be done so that it may be published in this journal. I have the following comments and recommendations.

- The introduction of relevant background and research progress was not comprehensive enough

-The purity of the raw materials used must be provided

-More details on pH choice should be provided. What motivated the authors for this investigation as a function of pH. Why is pH important for this study?

-- There is evidence that pH, in addition to morphology, alters the structural properties of materials. Evidently, it is unreasonable to treat an application without a clear understanding of the structure and the parameters that modify it. Thus, the effect of pH on the crystal structure of the material under analysis should be investigated and provided in this work in more detail.

- About figure 3 there are several incongruous things. 1) The caption of the figure does not correspond with the images presented. 2) Figure 3a does not correspond to a ZnO diffraction pattern. This should be reviewed and corrected. Authors should be more careful with the information provided.

-The SEM images are at different magnifications, which does not allow a comparison on the effect of pH. Images in the same magnifications must be provided.

- Why do EDS show peaks of Cu and other elements, this should be discussed

-The quality of the figures in general are bad, they should be improved

- It is appreciable that the authors studied: Synthesis, and Characterization, of Zinc Oxide nanoparticles at different pH values from Clinopodium vulgare L. and their assessment as an antimicrobial agent and biomedical application, however, the work became descriptive and qualitative. An in-depth scientific discussion of the main results obtained should be carried out.

Author Response

Reviewer 2

Query: The introduction of relevant background and research progress was not comprehensive enough

Response: Improved as per suggestion.

Query: The purity of the raw materials used must be provided

Response: Added on page 2.

Query: More details on pH choice should be provided. What motivated the authors for this investigation as a function of pH. Why is pH important for this study?

Response: pH is a very important factor for any chemical reaction. Change in pH can affect the synthesis of nanoparticles by causing agglutination. For more detail the Importance of pH has been added in Introduction section.

Query: There is evidence that pH, in addition to morphology, alters the structural properties of materials. Evidently, it is unreasonable to treat an application without a clear understanding of the structure and the parameters that modify it. Thus, the effect of pH on the crystal structure of the material under analysis should be investigated and provided in this work in more detail.

Response: For better understanding and detail the Importance of pH has been added in Introduction section

Query: About figure 3 there are several incongruous things. 1) The caption of the figure does not correspond with the images presented. 2) Figure 3a does not correspond to a ZnO diffraction pattern. This should be reviewed and corrected. Authors should be more careful with the information provided.

Response: Suggested changes has been done.

Query: The SEM images are at different magnifications, which does not allow a comparison on the effect of pH. Images in the same magnifications must be provided.

Response: Sir we used SEM analysis for the study of surface properties. We used SEM images are at different magnifications because if we use the images of same magnification the images were not clear so that’s why images of different magnification. Further for comparative analysis we used different characterization like XRD, UV, FTIR and EDX which clearly shows the differences.  

Query: Why do EDS show peaks of Cu and other elements, this should be discussed

Response: The presence of Cu in EDX confirms nanoparticle preparation via the green synthesis method. The Cu may come from the plant extract which was present as a raw material.

Round 2

Reviewer 1 Report

The authors addressed all comments. 

It is ok. 

Reviewer 2 Report

This version can be accepted for publication.